# Research on Data Link Channel Decoding Optimization Scheme for Drone Power Inspection Scenarios

Haizhi Yu [1], Kaisa Zhang [2], Xu Zhao [3], Yubing Zhang [3], Bingfeng Cui [4], Shujuan Sun [5], Gengshuo Liu [5], Bo Yu [6], Chao Ma [6], Ying Liu [6] and Weidong Gao [1,*]

[1] School of Information and Communication Engineering, Beijing University of Posts and Telecommunications, Beijing 100876, China; yhz_study@bupt.edu.cn

[2] School of Electronic Engineering, Beijing University of Posts and Telecommunications, Beijing 100876, China; kaisa@bupt.edu.cn

[3] Beijing Smartchip Microelectronics Technology Company Limited, Beijing 100089, China; zhao@sgchip.sgcc.com.cn (X.Z.); zhangyubing@sgchip.sgcc.com.cn (Y.Z.)

[4] State Grid Corporation of China, Beijing 100031, China; bfcui@sgcc.com.cn

[5] State Grid Xiongan New Area Electric Power Supply Company, Baoding 071600, China; xa_sunsj@he.sgcc.com.cn (S.S.); xa_liugs@he.sgcc.com.cn (G.L.)

[6] Yingli Energy Development Co., Ltd., Baoding 071000, China; vincent.yu@yingli.com (B.Y.); machao@yingli.com (C.M.); ying.liu@yingli.com (Y.L.)

[*] Correspondence: gaoweidong@bupt.edu.cn

**Abstract:** With the rapid development of smart grids, the deployment number of transmission lines has significantly increased, posing significant challenges to the detection and maintenance of power facilities. Unmanned aerial vehicles (UAVs) have become a common means of power inspection. In the context of drone power inspection, drone clusters are used as relays for long-distance communication to expand the communication range and achieve data transmission between patrol drones and base stations. Most of the communication occurs in the air-to-air channel between UAVs, which requires high reliability of communication between drone relays. Therefore, the main focus of this paper is on decoding schemes for drone air-to-air channels. Given the limited computing resources and battery capacity of a drone, as well as the large amount of power data that needs to be transmitted between drone relays, this paper aims to design a high-accuracy and low-complexity decoder for LDPC long-code decoding. We propose a novel shared-parameter neural-network-normalized minimum sum decoding algorithm based on codebook quantization, applying deep learning to traditional LDPC decoding methods. In order to achieve high decoding performance while reducing complexity, this scheme utilizes codebook-based weight quantization and parameter sharing methods to improve the neural-network-normalized minimum sum (NNMS) decoding algorithm. Simulation experimental results show that the proposed method has a better BER performance and low computational complexity. Therefore, the LDPC decoding algorithm designed effectively meets the drone characteristics and the high channel decoding performance requirements. This ensures efficient and reliable data transmission on the data link between drone relays.

**Keywords:** power inspection; drone relays; data link; deep learning; LDPC decoding

## 1. Introduction

In recent years, the rapid development of smart grid technology has propelled the construction of smart grids into a rapid growth stage. To meet the increasing demand for electricity, numerous new transmission lines have been deployed, resulting in a significant surge in transmission line mileage and the amount of power equipment. Moreover, many transmission lines are located in areas with complex terrain and harsh environments, which significantly increases the possibility of transmission line failures. As a result, the monitoring and maintenance of power facilities in electric power transmission are an

enormous challenge. At present, the traditional manual detection methods involve heavy workloads and high risks, rendering them inadequate to meet the detection requirements of the modern power industry. These limitations have created substantial difficulties in the monitoring and maintenance of equipment on large-scale transmission lines [1]. Nowadays, unmanned aerial vehicles (UAVs) have emerged as a promising solution for transmission line inspections, primarily due to their features such as beyond-line-of-sight flying, enhanced image quality, and ease of operation. By utilizing drones to detect transmission line faults, inspectors can effectively reduce high-risk tasks such as tower climbing, resulting in a substantial reduction in workload and risk. UAVs have been extensively employed in transmission line inspection, enabling more automated and efficient intelligent power inspections [2].

In the scenario of drone power inspection, due to the considerable distance between the inspection area of the working drone and the ground server or base station, there is the challenge of long-distance communication. With the maturing of drone technology, UAVs have become a new possibility for drones to be used as aerial base stations or mobile relay stations to solve long-distance communication problems [3]. UAV-aided hybrid communication techniques have been widely investigated [4], and wireless communication assisted by drones has proven effective in scenarios with widely distributed users and significant communication obstacles. The concept of drone relays has been extensively discussed [5,6]. By deploying specific drones as relay nodes, it becomes possible to amplify and forward signals, resulting in a significant expansion of the communication range, enabling wireless transmission over long distances [7]. The design of a drone relay system plays a crucial role in ensuring effective communication and coordination between the drones, enabling wireless relay functionality and efficient transmission of signal. Li, Guo, et al. [8] studied drone-cluster-assisted multi-hop communication schemes for scenarios where the distance between the source and destination nodes exceeded a certain threshold. This study explored the effectiveness of utilizing a cluster of drones to facilitate multi-hop communication. Chen, Zhao, et al. [9] discuss the optimal placement of multiple drones in two typical relay settings: establishing a single multi-hop link or multiple double-hop links. This study aims to specifically focus on the multi-hop single-link power inspection scenario involving multiple UAVs. This scenario serves as a solution to extend the signal transmission distance and expand communication coverage. By implementing a multi-hop single-link communication system model with multiple drone relays, the drones are capable of remotely transmitting power data captured from the work area to the ground server or base station, enabling efficient long-distance communication.

Wireless communication technology plays an important role in the above power inspection scenarios that rely on multiple drone relays. The drone data link serves as a crucial component of the drone system, and its reliability is a key factor affecting drone wireless communication. When drones perform inspection tasks in complex environments, they need to maintain real-time wireless connectivity with ground servers and drone relays, and transmit a large amount of data, such as captured power facility images, to ground servers. This requires the data link to have high transmission efficiency and reliability. Therefore, the primary focus lies in exploring methods and techniques that enable the efficient and reliable transmission in data links. In the drone cluster relay system, data communication between drones or between drones and ground servers is susceptible to channel interference and additive noise. In order to improve the reliability of the data link, channel encoding, also known as error control encoding, is usually used. This article aims to focus on the research of channel encoding and decoding technology for the data link between drones and drone relays.

Convolutional code, turbo code and low-density parity-check (LDPC) code are commonly used for data channel coding and decoding. In power inspection scenarios, some drones serve as relays for auxiliary communication, necessitating the encoding and decoding of a substantial amount of received power data. Due to the limited payload and computing power of drones, the channel encoding and decoding algorithms for drone

relays must have the characteristics of low complexity, fast computation speed, and high reliability. Therefore, a low-complexity and high-reliability encoding and decoding method for drone data links is needed. Research has shown that turbo codes exhibit excellent performance even under low signal-to-noise ratio conditions. Berrou et al. [10] point out turbo codes can achieve high decoding performance in the presence of Gaussian white noise interference. But turbo decoding algorithms are characterized by low throughput and complex implementation [11]. This limits their applicability in drone communication scenarios. LDPC codes not only possess strong error correction capabilities and are easy to perform parallel operations, but also are closest to the Shannon limit and have low decoding complexity [12,13]. A performance comparison of the three channel coding and decoding schemes is shown in Table 1.

**Table 1.** Performance comparison of commonly used channel encoding and decoding schemes.

| Performance | LDPC Code | Turbo Code | Convolutional Code |
|---|---|---|---|
| Error correction ability | Pretty strong | Pretty strong | Strong |
| Fragrance limit difference | 0.0045 dB | 0.5 dB | >4 dB |
| Encoding and decoding complexity | Medium | Complex | Simple |
| Anti-interference ability | Strong | Strong | Average |
| Throughput | >100 Mbps | 1–100 Mbps | >100 Mbps |
| Calculation | Parallel | Parallel | Parallel |

Therefore, LDPC codes are considered the optimal data channel coding scheme for UAV data links. Utilizing LDPC codes can enhance the reliability of data transmission in drone relay communication and improve the resistance to interference. In order to better meet the high reliability and low complexity requirements of unmanned aerial vehicle data link transmission, this study discusses LDPC decoding algorithms, with a focus on the trade-off between LDPC decoding performance and computational complexity.

LDPC codes were initially proposed by Gallager in 1962 [14]. The main LDPC decoding algorithms include belief propagation (BP) and min-sum (MS) algorithms. The BP decoding algorithm exhibits excellent performance but requires extensive logarithmic and multiplication operations, resulting in high computational complexity. In contrast, the MS decoding algorithm significantly reduces computational complexity at the expense of performance degradation. Furthermore, the MS algorithm can be improved by using correction factors to adjust the calculation results of the checking nodes, known as the normalized MS (NMS) algorithm or offset MS (OMS) algorithm [15,16]. These decoding algorithms can improve decoding performance and reduce complexity [17].

The field of UAV data link communication faces multiple technological challenges, including high-speed motion and Doppler effects, channel fading, signal interference and noise, limited bandwidth and spectrum scarcity, resource constraints, rapid deployment, and dynamic network topologies, as well as data transmission security and privacy protection. These challenges collectively constitute the complexity of ensuring reliability, performance, and security in UAV data transmission, necessitating innovative channel decoding algorithms and communication technologies to address them.

Addressing the requirements of simplicity, speed, and strong error correction capabilities for channel coding algorithms in UAV data link onboard terminals, we conducted an analysis of the coding characteristics suitable for UAV data link systems. Nurbani et al. [18] introduced a new family of polar codes with a hybrid multikernel tailored for short blocklength transmissions, which is well suited for communicating control data between UAVs. As previously mentioned, LDPC codes are better suited for long-code transmissions, meeting the requirement of transmitting a large amount of power long-code data between drones in the scenario described in this paper. Yu et al. [19] compared LDPC codes, convolutional codes, and turbo codes, with simulation results consistently demonstrating



the superior performance of LDPC codes. This makes them a preferred channel coding solution for UAV data link systems. In the context of UAV cluster data exchange, Wang et al. [20] applied LDPC codes for error correction and improved the decoding algorithm through log-likelihood ratio (LLR) calculations based on the max-log-MAP criterion. Simulation results have shown its ability to meet the strong interference resistance and high channel utilization requirements, making it suitable for UAV cluster data links. Additionally, Wang et al. [21] introduced a CEC-LDPC coding method, which was found to be better suited for UAV data link downlink applications. Zhu et al. [22] proposed an unmanned aerial vehicle telemetry and control transmission system based on Q-LDPC codes, and their simulation results demonstrated that this decoding scheme effectively combats fast fading, reducing the bit error rate and providing safety protection for UAV ground observations by converting continuous burst bit errors into a smaller number of symbol errors. However, it is worth noting that the complexity of Q-LDPC code decoding is relatively high, making it less suitable for resource-constrained UAV devices. The technologies mentioned thus far may face challenges related to high decoding complexity or relatively low decoding efficiency, which may not be suitable for the system model discussed in this paper. In comparison to previous research, our proposed enhanced LDPC-SNNMS decoding scheme introduces machine learning techniques into UAV data link channel decoding, leading to significant improvements in performance and reliability, and adaptability to the resource limitations of UAVs.

Due to the high requirement for decoding performance in drone communication scenarios to ensure the reliability of the data link, traditional LDPC decoding algorithms may not be able to meet the desired performance. Considering that UAVs possess powerful computational capabilities and embedded systems, enabling them to execute complex algorithms, this study combines machine learning with traditional LDPC decoding algorithms to enhance decoding performance and meet the high reliability requirements of data transmission between unmanned aerial vehicles. Considering the limited computational resources and battery capacity of drones, this study focuses on the design of a deep-learning-based LDPC decoding algorithm that can achieve high performance while minimizing computational complexity, storage space, and energy consumption.

There are two methods in deep learning: model-driven methods and data-driven methods. Liang et al. [23] proposed an iterative BP-CNN architecture that connects a trained CNN with a BP decoder to improve the BER performance. Gruber et al. [24] addressed short codes and introduced a deep-learning-based decoder that improves BER performance through training on a large number of codewords. Zhang et al. [25] combined the traditional BP algorithm with the FGNN algorithm [26]. These data-driven methods show excellent training performance for short codewords. In the context of drone power inspection scenarios, the collected data primarily consist of long code, such as power facility images. The data-driven approach relies on feeding a large amount of data to train neural network models, which results in high computational complexity and increased resource consumption. This makes the data-driven methods less suitable for channel decoding algorithms applied to drone scenarios. However, the model-driven methods can effectively address this issue. In [27,28], a model-driven neural network BP decoding was proposed, which achieves decoding performance similar to or higher than traditional BP algorithms with fewer iterations. Abotabl et al. [29] introduced a model-driven learning offset minsum algorithm framework, which exhibits low complexity and can be applied to any decoding schedule. Xu et al. [30] presented a model-driven polar decoding algorithm that effectively balances complexity and performance. Wang et al. [31] proposed a model-driven normalized min-sum (NMS) decoding method, which requires fewer iterations compared to traditional LDPC decoding and achieves improved decoding performance. Therefore, to minimize computational complexity and resource consumption while ensuring BER performance, this study designs a model-driven LDPC decoding algorithm, aiming to reduce decoding complexity by reducing the number of iterations or operations. Although the method reduces decoding complexity to some extent, the presence of a large number of

weights in the neural network still results in high computational complexity. In the context of drone power inspection, further improvements are needed to make it more suitable for practical applications.

To address the issue of high decoding complexity caused by multiple weight parameters, eliminating redundant parameters is crucial. Wang et al. [32] mentioned a neural two-dimensional normalized minimum sum decoder with a simplified parameter set, which allows for assigning the same weight to a class of similar messages. Lian et al. [33] introduced a parameter sharing scheme in the weighted belief propagation [27] decoder, where certain edges share the same weights to reduce storage and computational burden. In [31,34], a shared neural normalization min-sum (SNNMS) decoding network was presented, where all edges share the same weights in each iteration, reducing the number of correction factors. It achieves an improved bit error rate (BER) with fewer trainable parameters compared to NNMS. Han et al. [35] mentioned training quantization and weight sharing methods using codebook-based weight quantization for weight sharing, resulting in deep neural network compression. Building upon [35], Qingle et al. [36] proposed a novel shared offset min-sum (SNOMS) approach, reducing the number of network weights through parameter sharing, with comparable decoding performance to NOMS, but significantly reduced computational complexity. Teng et al. [37] proposed a low-complexity recursive neural network (RNN) polar decoder based on codebook weight quantization to reduce the weight parameters. This method effectively reduces memory overhead by 98%. Parameter sharing methods effectively address the issue of excessive learnable weights in the network, thus reducing the implementation complexity of the decoding algorithm while maintaining decoding performance.

The proposed improvement to the LDPC-SNMMS decoding scheme, achieved through extensive training and real-time data feedback, enables automatic adjustments to its internal parameters, allowing for better adaptation to continuously changing channel conditions. It maintains outstanding performance even in complex multi-path propagation and high-speed motion scenarios, such as those encountered by drones. Furthermore, this decoding scheme offers lower computational complexity, meeting the resource constraints of UAVs. Deep learning neural networks can be highly optimized for hardware to ensure efficient execution on embedded UAV computing platforms, minimizing resource utilization and thereby extending flight times and enhancing mission efficiency. The innovative contribution lies in harnessing the potent capabilities of machine learning and applying them to the UAV data link domain. This novel approach introduces deep learning neural networks that combine mathematical foundations and key concepts with machine learning to achieve adaptive channel decoding. This innovative method addresses some limitations of previous solutions.

Due to the large amount of power data transmitted between drones, to design a high-accuracy and low-complexity decoder for LDPC long-code decoding this study proposes a shared-parameter neural-network-normalized minimum sum algorithm based on codebook quantization (shared-NNMS-CQ).

The main contributions of the paper can be summarized as follows:

- Firstly, we incorporate the concept of the model-driven approach and leverage the advantages of neural networks and Tanner graphs to expand the iterative decoding process of checking the update and propagation of messages between nodes and variable nodes in the minimum sum algorithm (NMS) into a deep feed-forward neural network. We use a novel method of sharing network parameters to improve the NNMS network, which is distinct from the shared-parameter method proposed in [31]. This approach improves the BER performance by reasonably reducing network parameters.
- Furthermore, to further improve the novel LDPC decoding algorithm, we introduce a weight quantization method based on a codebook. This approach not only reduces the precision of each weight but also decreases the number of weight types required for training the network, leading to reduced computational complexity and resource

consumption. The reasonable selection of the quantization schemes can even have a positive impact on the decoding performance.

- Finally, to validate the BER performance of the proposed improved LDPC decoding algorithm, we demonstrate a performance and complexity analysis under a Rician channel in the context of drone communication scenarios.

## 2. Materials and Methods

### 2.1. System Model

In the context of drone power inspection, the drone inspection area is far from the ground server (or base station), resulting in weak data transmission. Moreover, the transmission performance of the drone data link is constrained by wireless channels. As shown in Figure 1, This study adopts a multi-hop single-link model based on the drone cluster relay system. In this model, electricity-related data needs to pass through multiple relay nodes from the source node to the target node, but each relay node still uses a single link for bidirectional data transmission. Because this article focuses on channel decoding algorithms, researching efficient and reliable channel decoding algorithms helps ensure reliable data transmission. The drone inspection area and base station (BS) serve as the source node and destination node, respectively [9]. This only discusses the channel decoding process of any one-way data transmission and will not affect our research content.

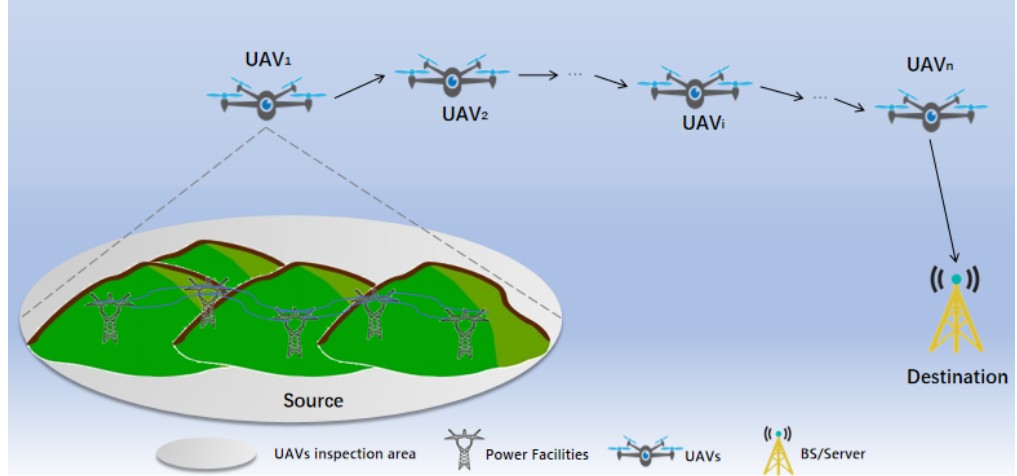

**Figure 1.** Multi-hop single-link system model based on the drone cluster relay.

The data link is divided into three parts: the ground-to-air uplink from the source node to UAV1, the air-to-air link from UAV1 to UAV2, and the air-to-ground downlink from UAVn to the target node. For multi-hop single-link settings, drones serve as communication relays to expand the communication range. The drone cluster as a data transmission relay provides an application background for our research. The focus of this article is to study channel decoding algorithms for data transmission between any two drones. Most communication occurs in the air-to-air channel between different drones. Therefore, to ensure the reliable transmission of collected data, it is necessary to focus on discussing the data transmission performance of the air-to-air link between drones. The air-to-air link between adjacent drones has direct and non-direct paths, so the air-to-air link channel is considered a small-scale fading Rician channel. This paper takes the BER performance and computational complexity as the optimization objectives of the system, aiming to achieve a balance between the optimal performance of the data link channel decoding algorithms and resource utilization. This will help improve the overall performance of the drone power inspection system, thereby ensuring reliable data transmission. The first objective is to ensure dependable data transmission, meeting the stringent requirements for data integrity and accuracy in power inspection scenarios. The second objective is to reduce

computational complexity, allowing the system to adapt to the limited resources available on drones.

### 2.2. Air-to-Air Data Transmission Link Model

The large amount of collected power data needs to be efficiently and reliably transmitted between the drone and the drone relay, or between the drone relay executing inspection tasks. Therefore, the drone channel adopts LDPC channel encoding and decoding to improve the anti-interference ability of the drone data link. The drone data link transmission model based on LDPC code is shown in Figure 2. $UAV_i$ ($i = 1, 2, 3, ..., n − 1$) serves as the transmitter, taking captured images, videos, and other power data as input. The input data are channel encoded using LDPC code. Then, BPSK digital modulation is used to map the binary data into constellation symbols, which are transmitted through the communication channel between unmanned aerial vehicles (i.e., the small-scale fading Rician channel of the air-to-air link). $UAV_{i+1}$ serves as the receiver, performing the opposite operation on the drone relay compared to the transmitter, namely, digital demodulation, LDPC decoding, and outputting data.

Within the data link system transmission model, this article focuses on the design of the LDPC decoder. The aim is to design a channel decoding algorithm that satisfies the characteristics of unmanned aerial vehicles and fulfills the high-reliability-communication requirements of air-to-air links.

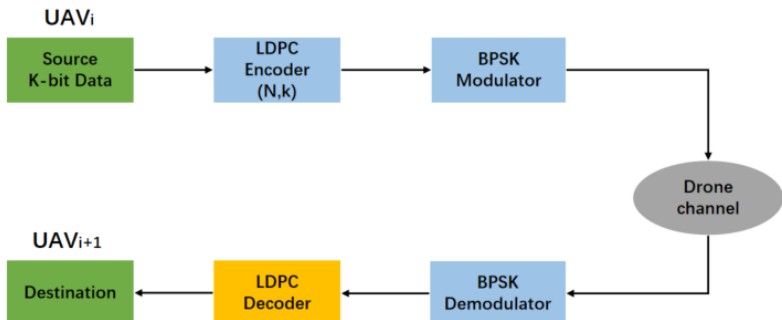

**Figure 2.** Air-to-air data transmission link model.

### 2.3. Prior Work

This section introduces the theoretical basis of LDPC encoding and decoding, the minimum sum decoding algorithm, the model-driven deep learning method, and the improved methods of shared parameters and codebook-based weight quantization. Our proposed LDPC long-code decoding networks are based on these contents.

LDPC codes (N,k) consist of k information bits and (N–k) parity bits. They offer advantages such as low decoding complexity and parallel decoding capability while approaching the Shannon limit. It is suitable for application to unmanned aerial vehicle communication. The parity-check matrix of LDPC [38] exhibits strong sparsity, containing only a few non-zero elements. The information is encoded using an LDPC encoder [39] to obtain codewords, denoted as y = x · G, where G is the generation matrix. When codewords pass through the channel, they are affected by noise. The decoder takes the maximum likelihood ratio of the signal affected by noise as its input. Through the continuous updating of the check matrix and variable matrix, that is, the iterative process of variable nodes (VNs) and check nodes (CNs) constantly updating and transmitting messages to each other, the LDPC iterative decoding algorithm realizes the channel decoding process. The flow of the LDPC iterative decoding algorithm is shown in Figure 3.

The LDPC iterative decoding algorithm includes the belief propagation (BP) algorithm, the min-sum (MS) algorithm, etc. While the BP algorithm exhibits excellent performance, it comes with high complexity. On the other hand, the MS algorithm simplifies the expression for updating check node information, reducing the complexity of the decoding algorithm.

However, it sacrifices some performance. In this paper, we improve the min-sum decoding algorithm to enhance its decoding performance while reducing complexity. This modification allows the decoding algorithm to meet the requirements of UAVs and be deployed on UAVs, ensuring the reliability of communication data transmission in UAVs.

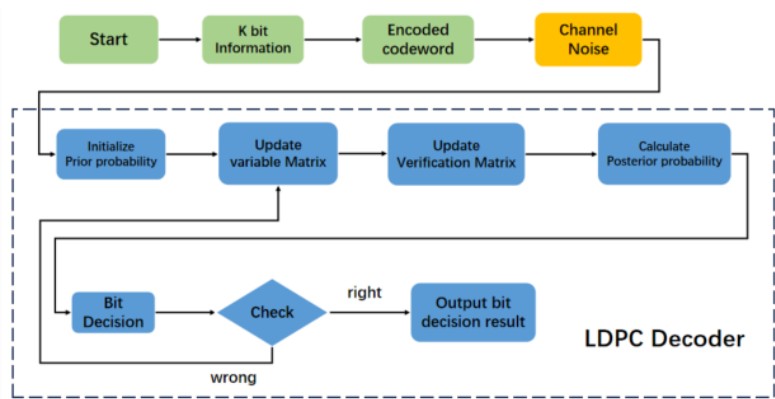

**Figure 3.** LDPC iterative decoding flowchart.

### 2.3.1. Normalized Minimum Sum Algorithm (NMS) and Model-Driven Methods

The main idea of the NMS algorithm is to adjust the calculation result by multiplying it with a correction factor during the step of updating the verification node information in the MS algorithm. The LDPC parity-check matrix H can be visually represented using a Tanner graph, where variable nodes (VNs) represent the bits in the codeword, and check nodes (CNs) represent the parity-check equations. In the LDPC parity-check matrix, H(i, j) = 1 indicates that the ith variable node (VN) is connected to the jth check node (CN) in the Tanner graph, while H(i, j) = 0 indicates no connection [40]. The LDPC decoding framework, as shown in Figure 4, is a bidirectional graph composed of check nodes, variable nodes, and the edges connecting them. The implementation of the NMS decoding process is represented on the Tanner diagram as iteratively updating and passing messages between VNs and CNs.

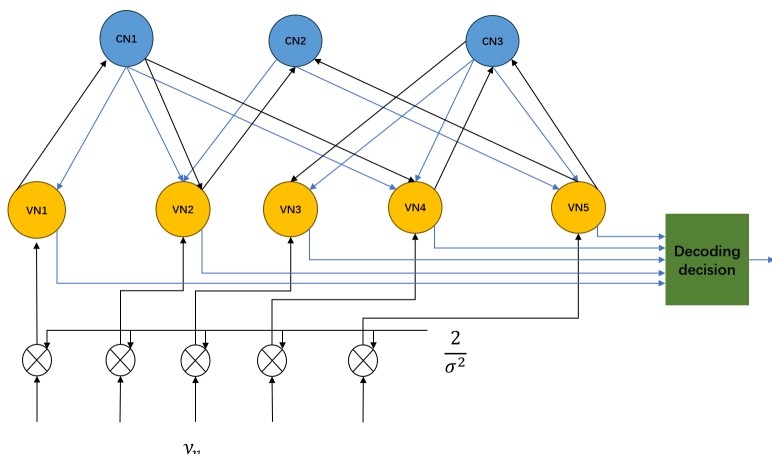

**Figure 4.** NMS LDPC decoding framework.

The Tanner graph is used to describe the structure of the codeword and the parity-check equations. If there is a connection between a VN and a CN in the Tanner graph, it indicates that the corresponding bit participates in the parity-check calculation. In the iterative update process of message passing, VNs update their log-likelihood ratio (LLR) based on the received messages and then send the messages to the connected CNs. CNs calculate the parity equations based on the received messages and send back the updated

messages to the connected VNs. By iteratively passing messages, errors can be corrected, and the accuracy of decoding can be improved.

Considering that the NMS algorithm reduces complexity at the cost of performance degradation, there is a need to enhance its performance while maintaining a relatively unchanged computational complexity in order to better apply it to the long-code decoding application of large amounts of drone data. Due to the successful combination of deep learning (DL) with many physical layer communication algorithms [41], the NMS algorithm is combined with DL techniques. DL can be divided into two methods: the data-driven methods and the model-driven methods. On the one hand, the data-driven methods require a large amount of training data to improve performance at the cost of significant latency and significant resource consumption. On the other hand, the model-driven methods combine communication knowledge with deep learning, based on prior knowledge of specific models and decoding algorithms, making neural networks interpretable to a certain extent, reducing the need for a large amount of training data, computational overhead, and training time [42]. Therefore, the NMS algorithm is expanded into a deep feed-forward neural network based on the Tanner graph structure, and the neural-network-normalized minimum sum (NNMS) decoding network based on the NMS decoding algorithm and model-driven deep learning network is obtained. The NNMS decoding algorithm can meet the high reliability requirements of drone communication.

In the air-to-air data transmission link model, binary code and binary phase shift keying (BPSK) modulation are used, and message bits are transmitted through the Rice small-scale fading channel [43]. The approximate decoding process of the NNMS LDPC decoder in this transmission model is as follows: after the decoder receives the signal affected by channel noise, the received channel LLR is calculated using the received symbol y for initialization, that is,

$$L_v = \log\left(\frac{P(Y_v = y_v \mid X_v = -1)}{P(Y_v = y_v \mid X_v = +1)}\right) = \frac{2y_v}{\delta^2} \tag{1}$$

Then, it is fed into the deep forward-propagation neural network of the LDPC decoder where the updates of the CN-to-VN messages and VN-to-CN messages in the MS algorithms are implemented by the CN layer and VN layer of the hidden layer of the neural network, respectively. Finally, the transmitted information bits are estimated. The main calculation formula for the NNMS decoding algorithm is as follows [31]:

The CN layer calculates messages from the CN to the VN:

$$U_{cv}^l = \prod_{v' \in V_j \setminus i} \text{sgn}\left(U_{v'c}^{(l-1)}\right) \cdot \min_{v' \in V_j \setminus i}\left|a_{v'c}^{(l-1)} \times U_{v'c}^{(l-1)}\right| \tag{2}$$

The VN layer calculates messages from the VN to the CN:

$$U_{vc}^l = L_v + \sum_{c' \in C_i \setminus j}\left(b_{c'v}^{(l)} \times U_{c'v}^l\right) \tag{3}$$

In the NNMS LDPC decoder, neural networks are introduced to learn the rules for message updates. The iterative update transmission process of messages in the Tanner graph is mapped to the hidden layer of the neural network, so that the entire NNMS LDPC decoding structure is similar to a neural network. In each iteration of the training process, the messages between the CNs and VNs in the hidden layer are multiplied by different trainable parameters for training, which is equivalent to assigning different weights to each connection on the Tanner graph. Then, supervised learning is carried out using the known correct codewords, the trainable parameters of the neural network are adjusted through backpropagation, repeating the training iterations until the network's performance converges or reaches the predetermined training iteration count. This process aims to make the network's output approximate the correct decoding result, achieving more accurate decoding with fewer iterations.

2.3.2. Codebook-Based Quantization Method

In this study, the network weights and parameters of the model-driven neural network are set as float32 floating-point type. However, a large number of floating-point parameters can hinder the hardware implementation of the neural network decoder. To address this issue, quantizing the parameters in the deep learning network can effectively reduce system resource overhead and decoding latency. The quantization methods mentioned in [35,37,44] do not degrade the BER performance of the LDPC decoding algorithm while significantly reducing complexity. First, multiple connections are made to share the same weights, reducing the number of effective weights that need to be stored. Then, a discretized codebook is designed to fine-tune these shared weights. The codebook is typically a finite set where each codeword represents a discrete value to replace the weights in the neural network. In neural networks, codebook-based parameter quantization is a technique that discretizes the representation of the network weights and activation values. It can effectively reduce the storage requirements and computational complexity of neural networks, thereby improving the model's operational efficiency. These methods can effectively reduce the number of parameters and lower the computational complexity. To maintain high performance, the weights are quantized after each training iteration, and allow the network to learn the optimal quantized weights.

This method can be seen as performing two rounds of quantization to reduce the precision and quantity of the required weights. Specifically, the quantization process is as follows:

First round of quantization: Quantifying the decimal places of 32-bit floating-point parameters to q bits. Second round of quantization: Designing a weight quantization method based on a codebook. When the size of the codebook is c bits, we calculate the frequency of occurrence of each type of weight from the $2^q$ types of weights, and select the $2^c$ most frequently used weights from large to small as the codebook values. Then, the weights are quantized to the nearest value in the codebook, further reducing the number of weight types in the neural network and, thus, reducing computational complexity. An example of the quantization process (q = 2, c = 2) is illustrated in Figure 5.

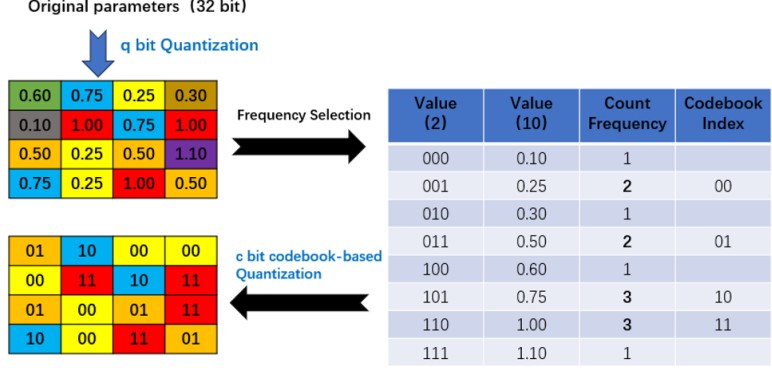

**Figure 5.** A weight quantization scheme based on a codebook.

By quantizing the network's weight parameters into a discrete codebook, the codebook-based weight quantization method can significantly reduce the storage requirements of the model. Moreover, the computational operations can be achieved through table lookup, reducing computational complexity. This makes it suitable for resource-constrained UAVs.

*2.4. The Proposed Shared-Parameter Neural-Network-Normalized Minimum Sum Decoding Algorithm Based on Codebook Quantization (Shared-NNMS-CQ)*

Building upon the aforementioned improvement methods, this study aims to design a high-precision, low-complexity LDPC long-code decoder. We propose a novel shared-parameter neural-network-normalized minimum sum decoding algorithm based on codebook quantization (shared-NNMS-CQ), which introduces innovative elements

including shared neural network learnable parameters and codebook-based weight parameter quantization. The specific improvements are outlined as follows:

- Shared neural network trainable parameters: A novel new shared-parameter NNMS (new-SNNMS) LDPC decoding is proposed. The weights $w_{cv}$ and $w_{vc}$ are independent and learnable. The weight vector $w_{cv}$ is shared across different iterations, and different weights are assigned to the check node messages in each iteration. The correction factor $w_{vc}$ is shared among all variable node messages in each iteration. This sharing mechanism reduces the number of correction factors and network weights, thereby reducing the computational complexity.

- Codebook-based weight parameter quantization: In the proposed new-SNNMS LDPC decoding neural network, the parameters in the network are quantized. First, the decimal places of float32 floating-point weights are quantized to q bits. Then, a codebook-based weight quantization method is used to reduce the number of weight types to $2^c$. This approach decreases the precision of each weight and reduces the variety of network weight values.

By incorporating these techniques into the NNMS LDPC decoding network, the shared neural network trainable parameters and codebook-based weight parameter quantization effectively reduce the computational complexity while maintaining decoding performance. The codebook-based quantization of the shared-parameter neural-network-normalized min-sum (shared-NNMS-Q) decoding network can be represented as a deep feed-forward neural network consisting of one input layer, L check node layers, L variable node layers, and one output layer. It comprises three types of neurons: neuron I (check node layer), neuron II (variable node layer), and neuron III (output layer). The deep-learning-based SNNMS decoding network, driven by a model, is illustrated in Figure 6.

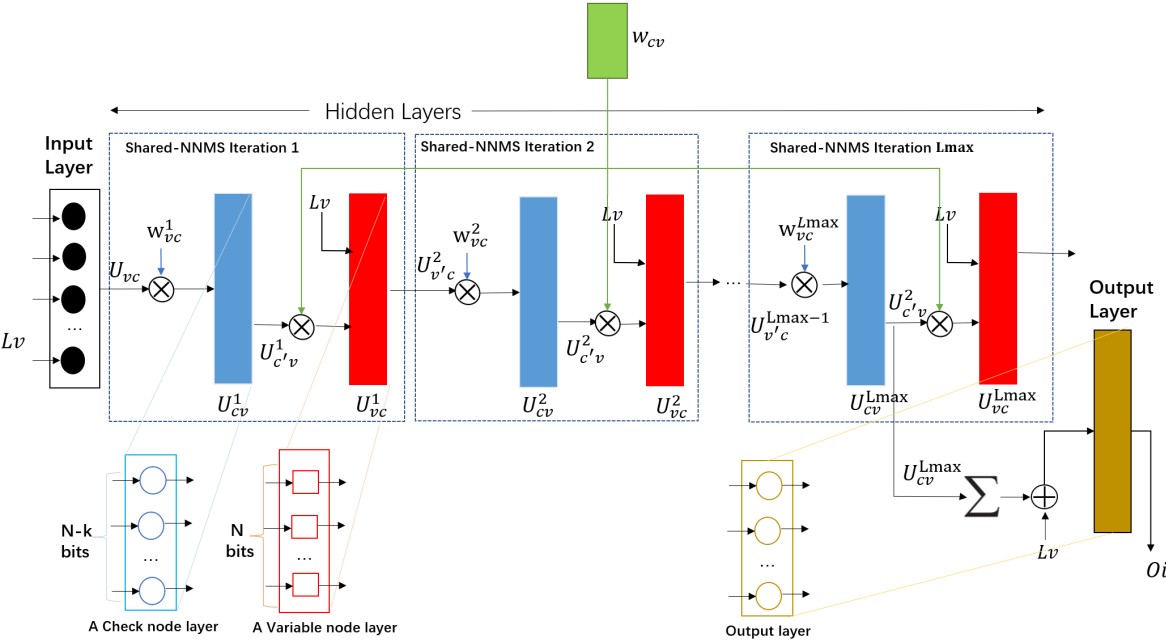

**Figure 6.** Multi-hop single-link model based on multi drone relay system.

We proposed the new-SNNMS algorithm, replacing Formulas (2) and (3) of the NNMS algorithm with Formulas (4) and (5) through the forward propagation of interpretable neural networks.

Neuron I (check node layer neuron) calculates the CN-to-VN messages in the CN layer:

$$U_{cv}^l = w_{vc}^l \times \prod_{v' \in V_j \setminus i} \mathrm{sgn}\left(U_{v'c}^{(l-1)}\right) \cdot \min_{v' \in V_j \setminus i} \left|U_{v'c}^{(l-1)}\right| \tag{4}$$

Neuron II (variable node layer neuron) calculates the VN-to-CN messages in the VN layer:

$$U_{vc}^{l} = L_v + \sum_{c' \in C_i \backslash j} \left( W_{cv} \times U_{c'v}^{l} \right) \tag{5}$$

Neuron III (output layer neuron) calculates the final output in the output layer:

$$o_i = \sigma \left( L_v + \sum_{c' \in C_i \backslash j} \left( W_{cv} \times U_{cv}^{(l_{\max})} \right) \right) \tag{6}$$

where L denotes the iteration index, $L_{max}$ represents the maximum number of iterations, and $\sigma(x)$ denotes the activation function.

Based on the above equations, it can be observed that the shared neural network with learnable weight vector $w_{cv}$ and correction factor $w_{vc}$ employs different sharing schemes, significantly reducing the number of weights and parameters in the network. While parameter sharing effectively reduces the required number of parameters, the use of float32 floating-point values still hinders the hardware implementation of the neural network decoder. Therefore, a codebook-based weight quantization method is applied. In the proposed shared normalized min-sum (shared NNMS) decoding neural network, which incorporates shared parameters, the training parameters of each layer undergo two quantization operations. Firstly, the precision of the weights and parameters is quantized. Then, based on a codebook, weight sharing is performed to further reduce the number of weight types in the network. This method effectively quantizes the weights and parameters in the network, reducing computational complexity and decoding latency without significant performance degradation.

## 3. Results

### 3.1. Experimental Setup

In this section, the Tensorflow framework [45] is used to train the decoding network. The parameter settings are shown in Table 2.

**Table 2.** Experimental Setup.

| Parameters | Values |
|---|---|
| Encoding | LDPC code (576,432) |
| Coding Rate | 3/4 |
| SNR | 5,6,7,8,9,10 |
| Batch Size | 240 |
| Optimizer | Adam |
| Learning Rate | 0.001 |
| Channel Model | Rician |

The parity-check matrix H of the LDPC code with a codeword length of 576 is selected according to the IEEE 802.16e standard [46]. The network is trained using mini-batch gradient descent, where each mini-batch consists of 240 data blocks generated at different SNR levels in the same proportion. The Adam optimizer is employed with a learning rate of 0.001 for optimizing the network parameters. The training data are generated based on [31] at multiple signal-to-noise ratios (SNRs), which ensures a more diverse training dataset and improves the BER performance of the network.

### 3.2. BER Performance vs. Number of Network Layers

Comparing the BER performance of new-SNNMS decoding networks with different numbers of layers, in Figure 7, we used 10, 12, 14, 16, 18, 20, 22, 32, and 42 layers.

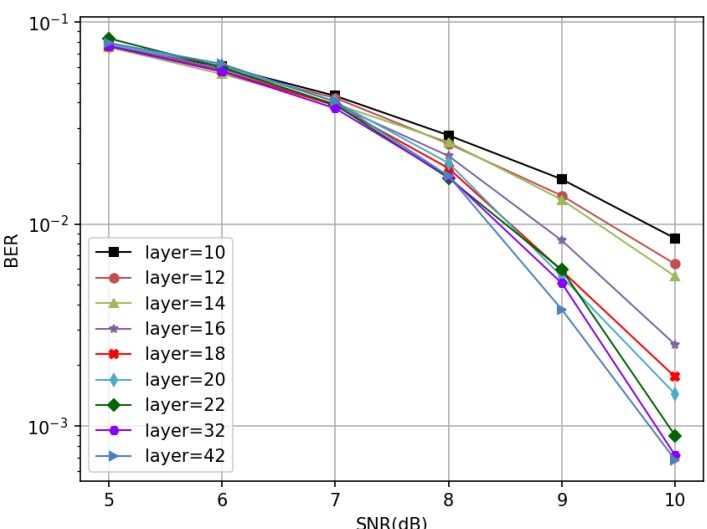

**Figure 7.** BER performance versus the number of network layers (new-SNNMS).

As shown in Figure 7, with the increase in the number of layers, the BER performance improves. The improvement in the decoding performance comes at the cost of increased computational complexity. Deeper network layers require more resources for additional multiplication and other operations. Therefore, while ensuring excellent BER performance, a reasonable number of network layers should be chosen to implement the decoding process.

As shown in Figure 8, when the number of neural network layers reaches a certain depth, the BER performance of the decoder will not significantly improve and gradually converge. Therefore, considering the trade-off between performance and complexity, we suggest choosing a 22-layer neural network.

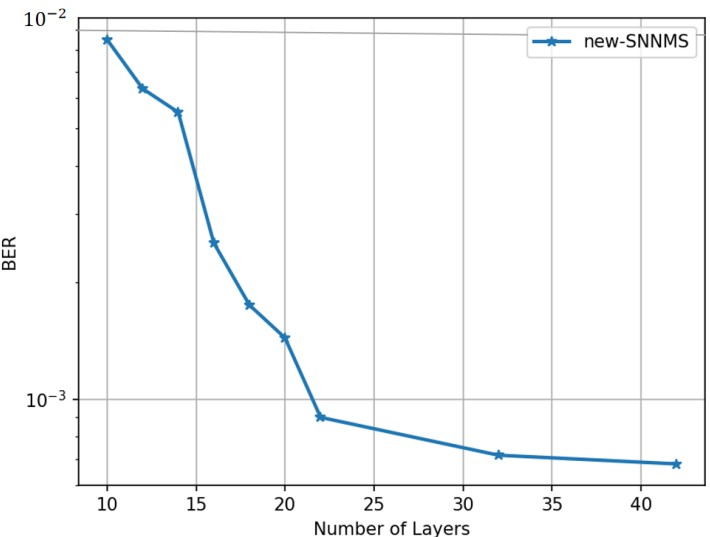

**Figure 8.** BER performance versus the number of network layers (SNR = 10).

### 3.3. BER Performance Comparison among NNMS, SNNMS, and New-SNNMS

Comparisons are made between the decoding performance of the proposed new-SNNMS decoding network and that of NNMS and NMS. Bellow, a 22-layer neural network is utilized for the comparison.

As shown in Figure 9, it can be observed that the BER performance of the new shared-parameter NNMS algorithm outperforms the SNNMS, NNMS, and NMS algorithms. In

the case of the 22-layer neural network (i.e., 10 iterations) with a BER of $10^{-3}$, the BER performance of new-SNNMS performs up to 0.3 dB better compared with the SNNMS.

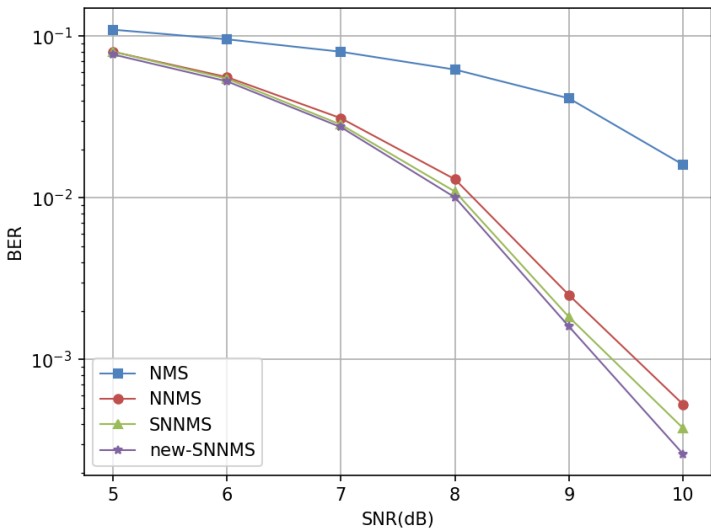

**Figure 9.** BER performance of different decoding schemes.

The new-SNNMS decoding algorithm exhibits better performance, it can achieve the same BER performance with fewer neural network layers. This implies that the decoding process exhibits lower latency and computational complexity. Therefore, it is suitable for decoding large amounts of power data transmitted on UAVs.

*3.4. Quantization*

Comparing the BER performance of different quantization schemes. The shared-NNMS decoding network with 22 layers (equivalent to 10 iterations) is employed to meet the hardware requirements for UAV flights by reducing system storage and computational complexity. To achieve this, a codebook-based quantization method is utilized. We evaluate the BER performance for various quantization parameters while considering the trade-off between performance and hardware consumption. In Figure 10, the codebook sizes range from 1 to 3, the quantization bits range from 3 to 5.

Based on the BER performance and storage space considerations, different quantization bits and codebook sizes are chosen. As shown in Figure 10, when the size of the codebook is c = 1, the BER performance deteriorates as the number of quantization bits increases. When the codebook size c is greater than 2, the greater the number of quantization bits, the better the BER performance. Furthermore, it can be seen that using the quantization scheme described in this article to quantify the weights of the NNMS decoding network, its BER performance may be even better than that of the unquantized BER performance.

In the context of drone power inspection scenarios, where reliable communication transmission between UAV relays is sought, careful selection of suitable weight types and precision is necessary to strike a balance between BER performance and computational efficiency. Through experimental evaluation, the optimal quantization parameter settings (c = 2, q = 4) can be determined to achieve reduced weight precision and types in the decoding network, enabling low-complexity and high-reliability communication transmission on UAV-to-UAV data links.

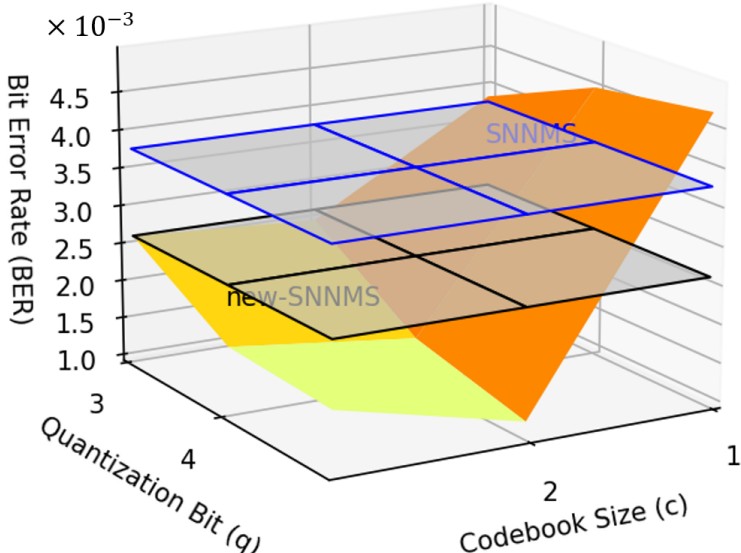

**Figure 10.** BER performance of shared-NNMS-CQ under different quantization settings.

*3.5. Complexity*

Comparing the computational complexity of different decoding algorithms. The complexity calculation includes comparison (CMP) operations, exclusive OR (XOR) operations, multiplication (MUL) operations, and addition (ADD) operations. The computational complexities are presented in Table 3, where T represents the number of iterations. The complexities of NMS, NNMS, and SNNMS are given by [26], their complexities are compared to the new-SNNMS that we proposed in this study.

**Table 3.** Complexity comparison of LDPC decoders.

| Item | NMS | NNMS | SNNMS | Shared-NNMS-CQ |
|------|-----|------|-------|----------------|
| CMP/XOR | $\sum_{j=1}^{N-k} a_j \left(a_j - 2\right) T$ | $\sum_{j=1}^{N-k} a_j \left(a_j - 2\right) T$ | $\sum_{j=1}^{N-k} a_j \left(a_j - 2\right) T$ | $\sum_{j=1}^{N-k} a_j \left(a_j - 2\right) T$ |
| ADD | $\sum_{i=1}^{N-k} (b_i(b_i - 1)T + 1)T$ | $\sum_{i=1}^{N-k} (b_i(b_i - 1) + 1)T$ | $\sum_{i=1}^{N-k} (b_i(b_i - 1) + 1)T$ | $\sum_{i=1}^{N-k} (b_i(b_i - 1) + 1)T$ |
| ML | T | $\sum_{j=1}^{N-k} a_j \left(a_j - 2\right) T + \sum_{i=1}^{N} b_i(b_i - 2)T$ | 2T | $\sum_{j=1}^{N-k} a_j \left(a_j - 2\right) T + T$ |

We can observe that the shared-NNMS-CQ decoder has lower complexity compared with the NNMS decoder. Compared to SNMS, the shared-NNMS-CQ achieves better BER performance, but the MUL operation of the shared-NNMS-CQ is slightly more complex than that of SNNMS. Furthermore, the shared-NNMS-CQ introduces weight quantization to reduce the number of training parameters and further decrease computational complexity. Overall, the improved decoding algorithm presented in this paper exhibits superior performance.

## 4. Discussion

In this paper, the proposed new-SNNMS algorithm exhibits superior performance compared to the SNNMS, NNMS, and NNMS algorithms. This may be attributed to the following points: Firstly, similar to the SNNMS decoding network, this decoding algorithm also maintains closer adherence to the original LDPC code structure in terms of its model mechanism. The proposed improved algorithm assigns different learnable parameters to the messages calculated by each variable node. This allows the algorithm to more flexibly adapt to complex data patterns and relationships, enabling finer parameter adjustments and, thus, enhancing the model's representation ability. Therefore, the model

exhibits higher accuracy and stronger generalization ability, resulting in more favorable training results.

By utilizing the proposed codebook-based weight quantization scheme to quantize the weights of our improved LDPC decoding network, we observed that this method even outperforms the unquantized BER performance. This may be because the quantization method can be seen as a regularization measure in the neural network. This method helps to reduce the risk of overfitting by reducing the representation ability of parameters, meaning that by reasonably reducing the type and accuracy of weights, the model is more inclined to learn simple and universal features. The effect of reducing overfitting may lead to better accuracy in test data, thus playing a positive role in improving performance. This scheme can effectively reduce the memory overhead and computational complexity of the proposed improved algorithm.

The experimental results have verified the applicability of our proposed LDPC decoding algorithm, which can be effectively applied to drones and improve the reliability of drone data link communication. This achievement in efficient and reliable communication transmission can bring numerous advantages for UAV power patrol, such as ensuring data integrity, enabling real-time monitoring and response, enhancing fault diagnosis and maintenance efficiency, and providing better security measures. And to a significant extent, it has significantly contributed to the real-time monitoring and response, fault diagnosis, and maintenance efficiency, as well as the overall safety of drone power inspection, thereby improving the operational efficiency of power companies, reducing maintenance costs, and improving the reliability and safety of power systems. Although the proposed decoding scheme demonstrates better BER performance, it still has relatively higher complexity compared to SNNMS. In future work, we aim to focus on complexity reduction and explore further improvement methods to achieve lower complexity and higher reliability in channel coding and decoding schemes, ensuring lighter and more reliable data transmission between UAVs. While the proposed decoding scheme exhibits better BER performance, it still demonstrates relatively higher complexity compared to SNNMS. In future work, our goal is to focus on reducing the complexity and exploring further improvement methods to design lighter and more reliable decoding schemes for unmanned aerial vehicle data link channels.

## 5. Conclusions

This paper explores the application of drones in power line inspection, addressing the challenges of detection and maintenance difficulties caused by the large-scale deployment of power facilities. It also proposes the use of drone clusters as data transmission relays to solve the problem of remote communication between patrol drones and ground servers. Due to the large amount of power data collected by drones, in order to improve the reliability of large-scale power data transmission on the drone data link, this study focuses on designing channel long-code decoding schemes for drone data links. LDPC codes are considered the best choice for decoding long codes in drone data links due to their excellent performance. By combining DL with the traditional LDPC normalized minimum sum (NMS) decoding algorithm, we propose a novel shared-parameter neural-network-normalized minimum sum decoding algorithm based on codebook quantization (shared-NNMS-CQ). This decoding algorithm introduces a shared-parameter method and weight quantization method to reduce computational complexity while ensuring decoding performance. Through experiments, it has been proven that the proposed method is a good decoding scheme for drone data link channel decoding.

**Author Contributions:** Conceptualization, H.Y.; methodology, H.Y. and K.Z.; software, H.Y.; validation, K.Z., X.Z., Y.Z., and B.C.; formal analysis, B.Y., Y.L., and C.M.; investigation, H.Y. and K.Z; writing—original draft preparation, H.Y.; writing—review and editing, W.G. and H.Y.; visualization, H.Y.; supervision, W.G., S.S., and G.L.; project administration, H.Y., K.Z., and W.G.; funding acquisition, S.S. and G.L. All authors have read and agreed to the published version of the manuscript.

**Funding:** The research in this article is supported in part by the State Grid Corporation Headquarters Science and Technology Project (Project Code: 5108-202218280A-2-410-XG) and in part by Beijing New Generation Information and Communication Technology Innovation Project (Project Code: Z231100005923026).

**Data Availability Statement:** Not applicable.

**Conflicts of Interest:** We declare no conflicts of interest with any individual or organization.

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
