# Peer review of "Research on Data Link Channel Decoding Optimization Scheme for Drone Power Inspection Scenarios"

_drones, doi:10.3390/drones7110662_

Round 1

Reviewer 1 Report

Comments and Suggestions for Authors

This is a well-structured article. Relevant reading with related references to extension to connecting nodes for extending communication range.

Comments on the Quality of English Language

Its good with minor errors example 

line 29  transmission faces   plural

33 Nowadays, unmanned  space

48 discussed [5,6].  space

Reviewer 2 Report

Comments and Suggestions for Authors

1. The introduction should be enhanced by highlighting the notable advancements and innovative contributions of this paper compared to prior research.
2. In the related works section, it's apparent that the authors allocate a significant portion of the text to introduce research in areas like UAV relaying, wireless communication, and coding. To enhance the paper, it is recommended that the authors focus on analyzing and comparing studies directly relevant to their research.
3. Although the authors have presented a system model, they have not provided a clear optimization objective.
4. The overall logical flow of the paper appears to need strengthening. The authors have dedicated a substantial part of the paper to describing previous research, while the analysis of the proposed algorithm is somewhat lacking.
5. The simulation results provided are rather limited and may not adequately demonstrate the superiority of the proposed algorithm. Further elaboration and experimentation could enhance the paper's credibility.        

Comments on the Quality of English Language

The narrative of the paper should be strengthened to enhance its logical coherence.

Reviewer 3 Report

Comments and Suggestions for Authors

The paper "Research on Data Link Channel Decoding Optimization Scheme for Drone Power Inspection Scenarios" proposes air-to-air channel decoding schemes for unmanned aerial vehicles (UAV) for power inspection. This paper proposes a high-accuracy, low-complexity decoder for LDPC long code decoding.

The system description (Section 2.1) assumes one-way data transmission. However, bidirectional transmission is necessary for typical systems used for power inspection (it may be asymmetrical).

The authors write that they "propose the use of drone clusters as data transmission relays." The article presents an outline of the system using the UAV chain. The problem of clustering has not been addressed here.

There are also editing errors in the article.
